EMBO
Molecular Medicine

# A universal SARS-CoV DNA vaccine inducing highly cross-reactive neutralizing antibodies and T cells

Sofia Appelberg[1],[†] (iD), Gustaf Ahlén[2,3],[†] (iD), Jingyi Yan[2,3],[†] (iD), Negin Nikouyan[2,3] (iD), Sofie Weber[4], Olivia Larsson[4], Urban Höglund[4], Soo Aleman[5,6], Friedemann Weber[7] (iD), Emma Perlhamre[8], Johanna Apro[8], Eva-Karin Gidlund[9], Ola Tuvesson[9] (iD), Simona Salati[10] (iD), Matteo Cadossi[10], Hanna Tegel[11] (iD), Sophia Hober[11], Lars Frelin[2,3], Ali Mirazimi[1,2,3,‡] (iD) & Matti Sällberg[2,3,*,‡] (iD)

## Abstract

New variants in the SARS-CoV-2 pandemic are more contagious (Alpha/Delta), evade neutralizing antibodies (Beta), or both (Omicron). This poses a challenge in vaccine development according to WHO. We designed a more universal SARS-CoV-2 DNA vaccine containing receptor-binding domain loops from the huCoV-19/WH01, the Alpha, and the Beta variants, combined with the membrane and nucleoproteins. The vaccine induced spike antibodies crossreactive between huCoV-19/WH01, Beta, and Delta spike proteins that neutralized huCoV-19/WH01, Beta, Delta, and Omicron virus *in vitro*. The vaccine primed nucleoprotein-specific T cells, unlike spike-specific T cells, recognized Bat-CoV sequences. The vaccine protected mice carrying the human ACE2 receptor against lethal infection with the SARS-CoV-2 Beta variant. Interestingly, priming of cross-reactive nucleoprotein-specific T cells alone was 60% protective, verifying observations from humans that T cells protect against lethal disease. This SARS-CoV vaccine induces a uniquely broad and functional immunity that adds to currently used vaccines.

**Keywords** DNA vaccine; *in vivo* electroporation; preclinical development; SARS-CoV-2; universal SARS vaccine
**Subject Categories** Immunology; Microbiology, Virology & Host Pathogen Interaction
**EMBO Mol Med (2022) e15821**

## Introduction

The SARS-CoV-2 pandemic has completely altered the way the society handles new viral infections, with lockdowns of cities or even entire countries (Smith *et al*, 2021). The extremely rapid development of vaccines by the scientific community and the pharmaceutical industry is an extraordinary achievement (Golob *et al*, 2021). A previous SARS-CoV-2 infection or a vaccination with either adeno-, DNA-, mRNA-, or protein-based COVID-19 vaccines induces neutralizing antibodies (NAbs; Varnaitė *et al*, 2020) and is effective in preventing symptomatic infection, and highly effective in preventing hospitalization or death (Chandrashekar *et al*, 2020; Feng *et al*, 2020; Guebre-Xabier *et al*, 2020; Mercado *et al*, 2020; van Doremalen *et al*, 2020; Yu *et al*, 2020; Lopez Bernal *et al*, 2021; Shaan Lakshmanappa *et al*, 2021; Vogel *et al*, 2021). However, the ability of the RNA genome of SARS-CoV-2 to undergo mutations and recombination poses continuous challenges (Miller *et al*, 2020). It has recently been shown that among the new mutant strains, especially the newly emerged variants of concern (VOC), anything from a few to multiple mutations in the receptor-binding domain (RBD) may render the current vaccines less effective against less severe breakthrough infections (Kustin *et al*, 2021; Wall *et al*, 2021). Several studies have highlighted that the Beta (B1.351), Delta (B.1.617.2), and Omicron (B.1.1.529) VOC may cause mild to moderate COVID-19 infections in those vaccinated (Madhi *et al*, 2021; Sadoff *et al*, 2021; Sheikh *et al*, 2021; Shinde *et al*, 2021). Recent data suggest that a previous infection with the original huCoV-19/WH01 wild-type strain or the Alpha (B.1.117) variant induces NAbs that retain strong cross-

1  Public Health Agency of Sweden, Solna, Sweden
2  Department of Laboratory Medicine, Karolinska Institutet, Huddinge, Sweden
3  Clinical Microbiology, Karolinska University Hospital, Huddinge, Sweden
4  Adlego AB, Uppsala, Sweden
5  Department of Infectious Disease, Karolinska University Hospital, Huddinge, Sweden
6  Department of Medicine Huddinge, Karolinska Institutet, Huddinge, Sweden
7  Institute for Virology, FB10-Veterinary Medicine, Justus-Liebing University Giessen, Giessen, Germany
8  Karolinska Trial Alliance, Karolinska University Hospital, Huddinge, Sweden
9  NorthX Biologics, Matfors, Sweden
10  IGEA Bomedical Spa, Carpi, Italy
11  Department of Protein Science, Royal Institute of Technology, Stockholm, Sweden
    *Corresponding author. Tel: +46 70 608 21 01; E-mail: matti.sallberg@ki.se
    †These authors contributed equally to this work
    ‡These authors contributed equally to this work

reactivity with the Delta (B.1.617.2) variant, whereas infections with the Beta (B.1.351) and Omicron (B.1.1.529) can escape these NAbs 12 (Liu et al, 2021). Thus, the appearance of mild to moderate breakthrough infections are not surprising. Fortunately, greater changes would most likely be needed to completely avoid the host immunity. Mutations in variants that escape NAbs appear in surface-exposed epitopic regions of the virus that are recognized more, or less, uniformly across humans. In contrast, the high diversity of the human leucocyte antigen class I and II makes it much less likely, or maybe even impossible, for T cell-escape variants to appear in viruses such as SARS-CoV-2 that are only causing acute infections (Alter et al, 2021; Tarke et al, 2021). Consistent with this, it has been found that vaccine mediated protection against severe disease from variants such as the Delta (B.1.617.2) variant is maintained, despite a reduced protection against mild and moderate disease (Sheikh et al, 2021). Consequently, although spike- and receptor-binding domain (RBD)-specific B cells may lack cross-reactivity, the spike-specific T cells are still cross-reactive toward the different variants (Alter et al, 2021; Tarke et al, 2021). This is further supported by the observation that T cells from SARS-infected patients who were infected with SARS-CoV in 2003 remain for 17 years and are to a large extent cross-reacting with SARS-CoV-2 (Le Bert et al, 2020). Overall, this strongly supports the notion that T cells may be able to confer a broader cross-reactivity than NAbs both between SARS-CoV-2 mutants and between different SARS viruses. Thus, new vaccines that combine sequences from the spike protein from multiple variants, combined with highly conserved viral protein sequences may overcome these problems.

## Results

Although SARS-CoV-2 can infect multiple species (Shi et al, 2020), it is clear that bats are reservoir for the origin of most human coronaviruses (Hu et al, 2015). The spike proteins of SARS-CoV and SARS-CoV-2 induce cross-reactive T cells, but poorly cross-reactive Nabs (Le Bert et al, 2020). Among the structural proteins of SARS-CoV and SARS-CoV-2, the envelope protein M and the nucleocapsid protein N have a higher genetic similarity to other animal SARS-CoV viruses than the receptor-binding S envelope protein (Hu et al, 2015; Latinne et al, 2020). T cells reactive to these two antigens show a higher cross-reactivity across betacoronaviruses (Ahlén et al, 2020; Le Bert et al, 2020). To take advantage of this, we designed a more universal SARS-CoV (OC-2.4) vaccine containing receptor-binding domain (RBD) loops of the S protein corresponding to the huCoV-19/WH01, Alpha, and Beta variants, combined with the M and N proteins of the huCoV-19/WH01 variant (OC2.4) (Dai et al, 2020; Fig 1A and Fig EV1). In particular, Beta variant share some of the spike mutations with the recently emerged Omicron variant BA.5, which supports the use of the Beta variant. An autoproteolytic P2A sequence was inserted between the RBD and the M and N proteins to avoid interference with the folding of the RBD. The M and N proteins were expressed as a fusion protein (Fig EV1B). Fig 1A illustrates the vaccine design and the concept of inducing both NAbs and broadly cross-reactive T cells. As control vaccines, we used either a pVAX plasmid without insert, or a recombinant S protein (huCoV-19/WH01) mixed with QS21 adjuvant. The

aim of a universal vaccine is to induce broadly reactive antibodies and T cells, that complements the COVID-19 vaccines already in use. As the S protein differs more and more between SARS-CoV strains and variants (Fig EV1), cross-reactive T cells targeting other regions may be of a growing importance.

We first immunized Balb/c mice with the OC-2.4 DNA using in vivo electroporation (EP) and found that they developed high levels of antibodies binding to recombinant S proteins of the huCoV-19/WH01, Beta, and Delta variants (Fig 1B, D, F and H) as well as to the N protein (mean $\pm$ SD anti-N endpoint titer: $25,920 \pm 28,979$). The S-specific antibodies effectively neutralized both the huCoV-19/WH01 and Beta variants of SARS-CoV-2 in vitro (Fig 1C and E).

In mice, priming with recombinant huCoV-19/WH01 S protein in adjuvant and boosting with the universal SARS-CoV DNA vaccine OC-2.4 effectively enhanced anti-S levels by 10- to100-fold to huCoV-19/WH01 S protein (Fig 1F). Importantly, heterologous boosting with the universal DNA vaccine OC-2.4 induced higher neutralization levels than homologous boosting (Fig 1G). Also, homologous boosting with the universal DNA vaccine OC-2.4 seemed superior in inducing NAbs to the Omicron variant (Fig 1G).

In rabbits, we found that one, two, and three doses of the OC-2.4 DNA vaccine induced high levels of anti-S antibody (Fig EV2) that effectively neutralized both Delta and Omicron variants of SARS-CoV-2 (Fig 1I). Thus, the inclusion of three RBD loops was effective at inducing broadly neutralizing antibodies to the huCoV-19/WH01, Beta, Delta, and Omicron variants. As a comparison, three doses of a recombinant spike protein in adjuvant corresponding to the huCoV-19/WH01 variant, was generally less effective in priming NAbs to the Beta, Delta, and Omicron variants (Fig 1).

Next, splenocytes (mice) and peripheral mononuclear cells (rabbits) from animals immunized with the universal SARS-CoV DNA vaccine OC-2.4 were analyzed for reactivity to S, M and M peptides and antigens, and cross reactivity of T cells to the Bat CoV N sequences (Fig 2). Mice develop T cells reactive to all components of the vaccine depending on the mouse strain, with Balb/c mice primarily developing T cells to S and N (Fig 2A–D). Importantly, the mean number of spot-forming cells (SFCs) per million splenocytes to both WH1 N and Bat N peptide pools number 3 (Fig 2), were significantly higher in the group that was primed with rS/QS21 and boosted with OC-2.4 DNA, as compared to their group only vaccinated with rS/QS21 (569 vs. 15, and 573 vs. 19, respectively, SFCs/ $10^6$ PBMC, $P < 0.0001$, Student's $t$-test, Graph Pad Prism). Thus, boosting with OC-2.4 DNA broadens the T cell reactivity in a host primed with a S-based vaccine. We could detect T cells to one or more components of the vaccine in PBMC from a majority of vaccinated rabbits at one or more time points, although some rabbits failed to develop detectable T cell responses at any time point (Fig 2E–H). In the vehicle group, none of the rabbits were reactive (> 50 SFCs/$10^6$ PBMCs) at days 14 (0/9) or 35 (0/11), whereas 1/11 at day 56 and 1/12 at day 77 were reactive to the N protein, as compared to 6/15 (40%; $P = 0.0519$, Fisher's exact test), 7/15 (47%; $P < 0.05$, Fisher's exact test), 9/13 (69%; $P < 0.01$, Fisher's exact test), and 10/18 (56%; $P < 0.05$, Fisher's exact test), respectively, at the same days in the group receiving 84 µg doses of OC-2.4 (Fig 2). The heterologous prime-boost strategy with priming with a spike-based vaccine and boosting with the OC-2.4 DNA vaccine effectively broadened the T cell response and introduced new T cell

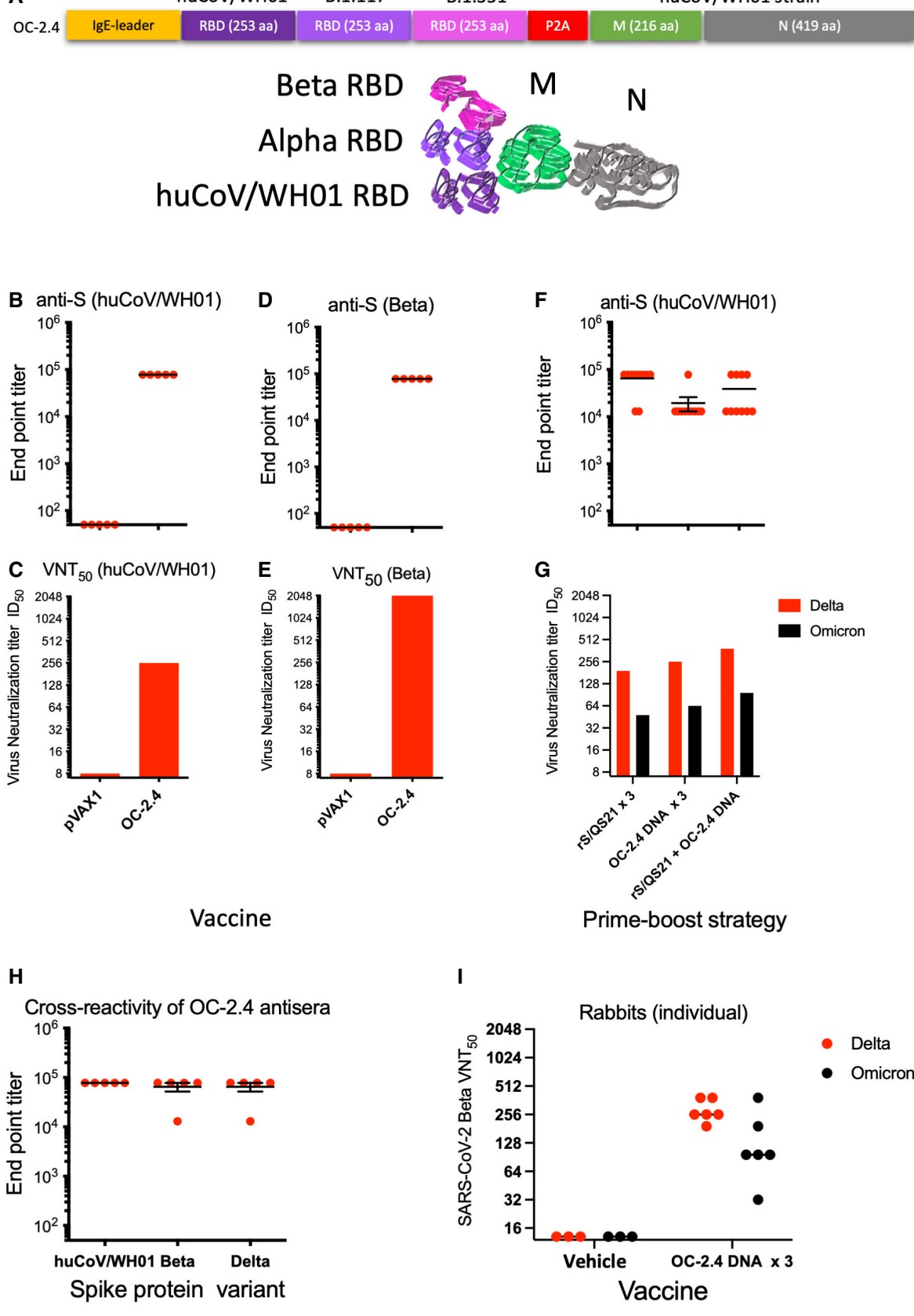

Figure 1.

**Figure 1. Immunogenicity of the universal SARS-CoV-2 vaccine.**

A    Schematic representation of the universal SARS-CoV-2 DNA vaccine OC-2.4 gene design has been given.

B–E   Ability of the SARS-CoV-2 DNA vaccine OC-2.4 to induce antibodies against S protein of the huCoV-19/WH01 (B) and Beta variants (D), and the ability to neutralize these viruses *in vitro* (C and E).

F, G   Also shown is the ability of the universal SARS-CoV-2 DNA vaccine OC-2.4 to prime and boost antibodies to S protein (huCoV-19/WH01) following priming with a recombinant S protein in adjuvant (F) and the ability of these antibodies to neutralize the Delta and Omicron variants *in vitro* (G).

H, I   Finally, three doses of the universal SARS-CoV-2 DNA vaccine OC-2.4 induce antibodies that cross react with S proteins from the huCoV-19/WH01, Beta, and Delta variants in mice (H), and four 840 μg doses of the same vaccine induces antibodies in female Zealand White rabbits that neutralize both the Delta and Omicron variants *in vitro* (I).

Data information: All samples were run individually in serial six-fold dilutions from 1:60 to 1:466,560 with groups of five to 10 mice (B, E, F, and H). In the NT assay, all serum samples from a group were pooled and serial two-fold dilutions from 1:16 to 1:4,096 were run in quadruplicate (C, E, and G). In (I), individual rabbit sera were run as serial two-fold dilutions from 1:16 to 1:4,096 in quadruplicate.

Source data are available online for this figure.

specificities to the spike-based vaccine (Fig 2A–D). This is a situation that is anticipated for the initial human use of this vaccine as a booster vaccine.

Importantly, the T cells reactive to the N protein were, unlike T cells reactive to the spike/RBD protein, highly cross-reactive to sequences corresponding to the Bat-CoV N in both mice and rabbits (Fig 2). Thus, these data shows that the vaccine induces highly cross-reactive T cells that also cross react with animal SARS-CoV proteins.

We also performed a full toxicological analysis of the rabbits following four doses of the vehicle or the OC-2.4 vaccine delivered by *in vivo* EP. At the injection site, degeneration and inflammation were seen 2 days after the last vaccination with vehicle for both the 84 and 840 μg dose (Table EV1). At 14 days after the last vaccination, these changes had healed completely in all DNA-vaccinated animals, except in one female that had a slight inflammation and degeneration (Table EV1). Also, no changes in biochemical or hematological markers could be linked to the treatments. Thus, apart from the OC-2.4 vaccine being immunogenic it also seems safe without any signs of pathology, except for immediately after vaccination at the treatment site, which is to be expected from a vaccine.

Finally, we analyzed the ability of the universal DNA vaccine OC-2.4 to induce protective immune responses against a lethal challenge with the SARS-CoV-2 Beta variant in human ACE2 transgenic K18 mice. Groups of mice were immunized three times, 3 weeks apart, and 2 weeks after the last dose the mice were challenged with $1 \times 10^5$ pfu of the SARS-CoV-2 Beta variant intranasally (Fig 3A), and then followed closely for symptoms and weight changes for 13 days. The universal DNA vaccine OC-2.4 fully protected these mice against

lethal infection and showed complete protection against viral replication in the upper airways and in the spleen (Fig 3C and D). However, these histopathological signs of disease were lower, albeit not significantly different from the mock-vaccinated group (Fig 3B). The viral replication in the lungs of the OC-2.4-vaccinated mice was also significantly reduced as compared to controls (Fig 3D). In contrast, vaccination with recombinant S in QS21 adjuvant also protected against lethal infection, and resulted in low infection in the lungs, but viral replication could still be detected in the upper airways and in the spleen (Fig 3C). Also this group did not show significantly lower histopathological signs of disease as compared to the mock-vaccinated group (Fig 3B). Interestingly, vaccination with an recombinant N protein (WH1 variant) in QS21 adjuvant (rN/QS21) that only activates antibodies and T cell responses to N, showed a 60% protection against lethal disease and viral replication in the upper airways, but less so in the lungs (Fig 3C and D). However, this group had a significantly more pronounced histological disease as compared to the groups vaccinated with recombinant S and OC-2.4 (Fig 2B). This strongly support the notion that T cells alone have a key role against protection against severe disease (Gao *et al*, 2022; Pardieck *et al*, 2022), and that the role of T cells may differ in their function in the upper and the lower airways.

To compare the priming of N-specific antibodies and IFNγ-producing T cells between DNA delivered by *in vivo* EP and a recombinant N protein in adjuvant, we immunized C57BL/6 mice (same background as the K18 mice) twice with either a plasmid encoding N alone, rN in alumn, or rN/QS21 (Fig EV3). A naïve group was included as negative control. This showed that rN/QS21 was most potent in inducing N antibodies, whereas DNA was most

**Figure 2. The universal SARS-CoV-2 vaccine induces broadly cross-reactive T cell responses.**

A–D   T cell responses as determined by ELISpot in Balb/c mice immunized three times with either spike protein in QS21 adjuvant (rS/QS21) (A), 50 μg universal SARS-CoV-2 vaccine OC-2.4 (B), primed with rS/QS21 and boosted with two doses of OC-2.4 (C), or empty pVAX plasmid (D), using *in vivo* electroporation. Spleens were harvested and individual mice were analyzed for the presence of IFNγ producing T cells using the indicated antigens. Data has been given as the number of IFNγ producing (spot forming) cells (SFCs) per million splenocytes. Each color indicate an individual mice.

E–H   Also shown is analysis of T cell responses in New Zealand White rabbits, which were immunized four times with either vehicle only (E), or 84 μg (F) or 840 μg (G) universal SARS-CoV-2 DNA vaccine OC-2.4 using *in vivo* electroporation. Peripheral blood mononuclear cells (PBMCs) were harvested and analyzed for presence of IFNγ producing T cells using the indicated antigens. Data has been given as the number of IFNγ SFCs per million PBMCs. The dotted line indicates the 50 SFC cut off. Also shown (H) is the cumulated IFNγ SFCs per million PBMCs to the RBD, M, and N peptide pools added together for each group of immunized rabbits (84 μg DNA, 840 μg DNA, and vehicle only) at each time point (day). The numbers on the *x*-axis of graphs (A–D) indicates the number of the respective peptide pool (WH1 variant) covering a part of the indicated protein.

Data information: All samples from mice and rabbits were run individually in triplicate and the mean number of SFCs/million have been given (A–G). In (H) the cumulated SFCs from all positive peptide pools from immunized rabbits have been given. The dotted line indicated the 50 SFCs/million cut off.

Source data are available online for this figure.

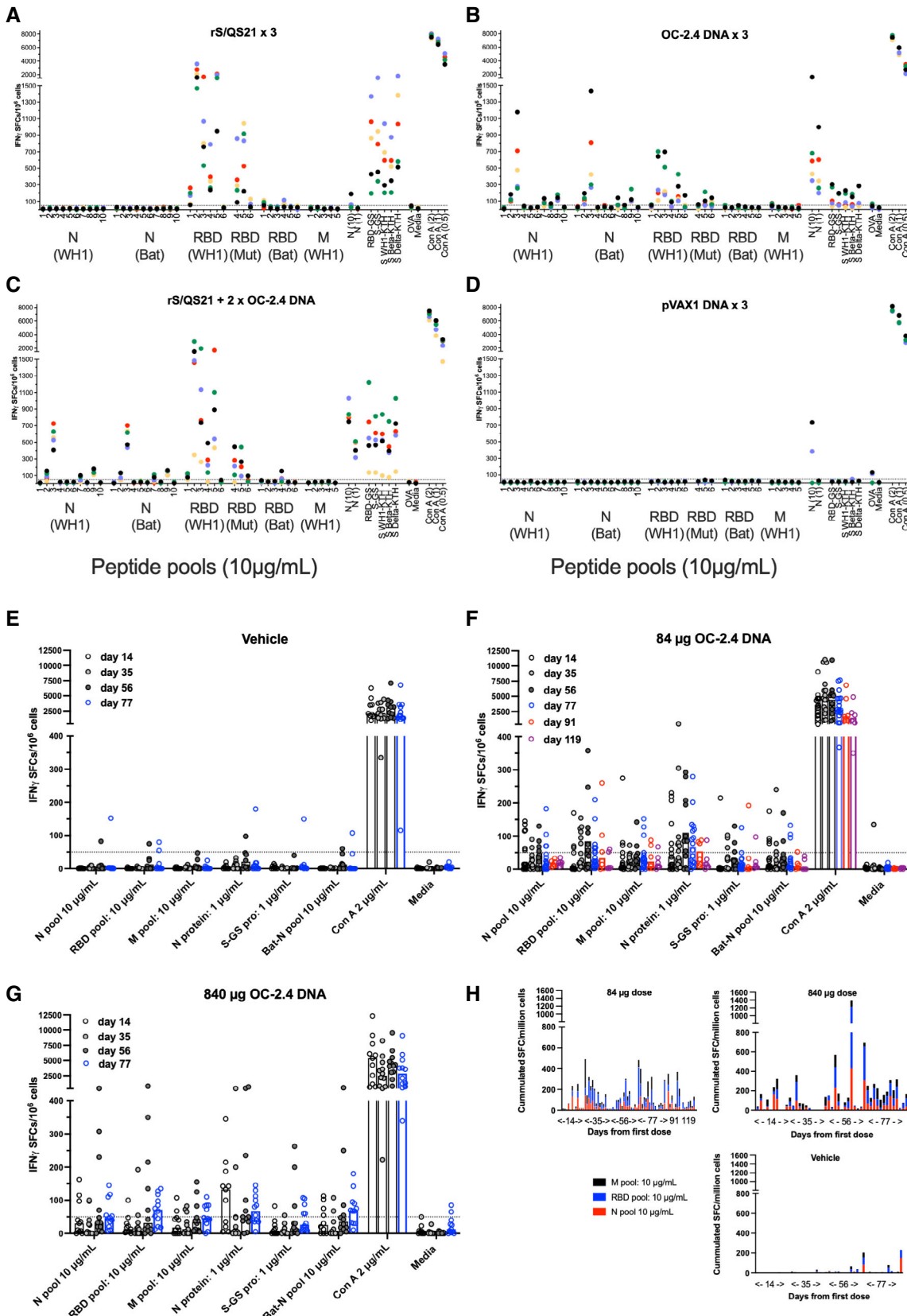

**Figure 2.**

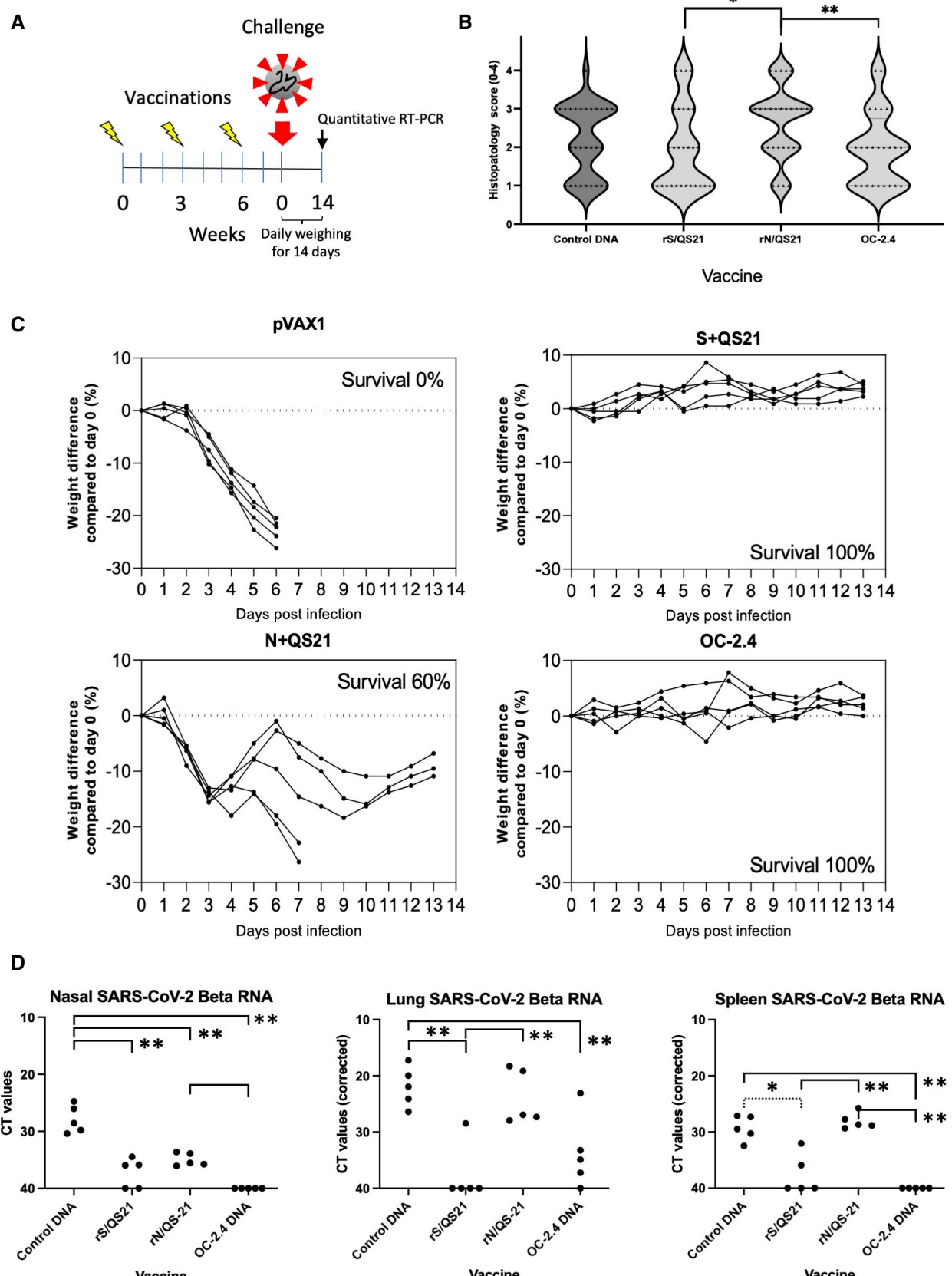

**Figure 3.**

**Figure 3.  The universal SARS-CoV-2 vaccine protects against K18 mice against lethal challenge with SARS-CoV-2 Beta variant.**

A      The experimental design of evaluation of different vaccine strategies in the K18-hACE2 mice transgenic for the human ACE2 receptor.

B, C    Three doses of respective vaccine fully or partially protected the mice against severe disease as determined by histological scoring of bronchial and alveolar lung tissues (B), percent weight loss (C).

D      Three doses of respective vaccine also protect against viral replication in the nose, lungs, and spleen. Values have been given as cycle times (CT), where lower values indicate a higher viral load. Statistical comparisons in the graph are shown with lines with one asterisk indicating $P < 0.05$ and two asterisks $P < 0.01$ (Mann–Whitney U-test, GraphPad Prism).

Data information: The histological scoring (B) was done by an independent pathologist unaware of the experimental groups. The data has been given as the individual histopathological score for each determination in each mouse ranging from 0 (none) to 4 (marked/severe) tissue damage. The weights were given as daily determinations of each mouse during the 14 study period (C). The levels of SARS-CoV-2 RNA in nasal washing has been given as the mean cycle time value from a duplicate determination of each sample. The levels of SARS-CoV-2 RNA in lung and spleen tissues has been given as the mean cycle time (CT) value from a duplicate determination. Each value was normalized by multiplying individual CT values with the following factor: the mean of all actin CT values divided by the sample actin CT value. Source data are available online for this figure.

effective in priming IFNγ-producing N-specific T cells, but comparable to rN/QS21 to priming IL-2-producing N-specific T cells (Fig EV3). Also, the reactivity to pool 6 peptides was most likely due to a CD8-restricted response, as it was only seen after immunization with DNA (Fig EV3). Overall, QS21 was superior to alumn in priming N-specific antibody and T cell responses (Fig EV3), which helps to explain its T-cell dependent and partially protective effect in the K18 model (Fig 3).

## Discussion

In this study, we herein designed and evaluated a completely new vaccine design strategy. We combined the RBD loops from three different variants of SARS-CoV-2, with the highly conserved M and N proteins. The concept was to induce broadly reactive and neutralizing antibodies, as well as broader and new T cell specificities as compared to any of the vaccines against SARS-CoV-2 used globally. We could show that when delivered as a DNA, the vaccine effectively induced broadly reactive and neutralizing antibodies as well as completely new T cell specificities. Importantly, we could show that this vaccine design could be used as a booster vaccine for the currently used spike-based vaccine, by improving anti-spike antibody levels, as well as broadening the T cell responses. Lastly, we could show that our vaccine effectively protected K18 hACE2 mice against a lethal disease caused by SARS-CoV-2 Beta variant, a variant sharing many mutations with the currently circulating Omicron variants. Thus, taking all data together with the favorable safety profile obtained in a toxicological study in rabbits, this supports a clinical development of this vaccine. A very interesting observation was that mice vaccinated with an N protein alone, despite being protected against lethal disease, had a more pronounced damage as compared to the mice with priming of both Nabs and T cells. This supports the concept that T cells indeed protect against severe disease and death (Pardieck *et al*, 2022). However, the absence of NAbs most likely allows for a viral spread and subsequent T cell-mediated killing resulting in a more pronounced histological disease. This requires further studies.

In conclusion, with a virus like SARS-CoV-2 that shows an impressive ability to mutate and to spread also among a population with high vaccine coverage, new vaccine designs are needed. Here, we describe a unique universal SARS-CoV DNA vaccine that induces more broadly neutralizing activity against the huCoV-19/WH01, Beta, Delta, and Omicron variants than standard spike-based vaccines. Our data supports clinical development of this completely new approach to vaccines against SARS-CoV-2 to complement existing vaccines as a heterologous booster, and potentially also protect against future SARS-CoV viruses that have replaced, or greatly changed the spike protein of SARS-CoV-2.

## Materials and Methods

### Animals

Female BALB/c (H-2$^d$) mice were obtained from Charles River Laboratories, Sulzfeld, Germany. Female B6.Cg-Tg(K18-ACE2)2Prlmn/J mice (JAX stock #034860) were purchased from Jackson Laboratory, USA. All mice were 8–12 weeks old at the start of the experiments and maintained under standard conditions at Preclinical laboratory (PKL), Karolinska University Hospital Huddinge, Sweden or Astrid Fagraeus Laboratory (AFL), Karolinska Institute, Solna, Sweden. Nine New Zealand White rabbits were purchased from Charles River, France and kept at AFL. All animal procedures were granted by regional animal ethics committees (approvals Dnr. 03634-2020, 17114-2020 and 16676-2020). Challenge experiments were performed in a biosafety level (BSL)-3 animal facility approved by the Swedish Board of Work Environment Safety.

### DNA plasmids and recombinant proteins

Vaccine candidate genes were generated based on the sequence from the huCoV-19/WH01 strain (Fig 1A). The genes contained the full Spike protein or a combination of the RBD, N, and M proteins, with a autoproteolytic P2A sequence. All sequences were codon optimized for expression in human cells and were synthesized by a commercial vendor (Genescript). Plasmids were grown in TOP10 *Escherichia coli* cells (Life Technologies) and purified for *in vivo* injections using Qiagen Endofree DNA purification kit (Qiagen) according to the manufacturer's instructions. The correct gene size was confirmed by restriction enzyme digests using BamHI and XbaI (Fast Digest; Thermo Fisher Scientific), and sequencing. Recombinant N protein was designed in-house and produced by Genescript (32878912; Ahlén *et al*, 2020). Recombinant S, RBD, and M was purchased from Genescript. The spike proteins were produced as full length by transient protein production in mammalian cells (Expi293). To facilitate trimerization of the full-length spike, a C-terminal T4 fibritin trimerization motif was included according to

Wrapp et al (2020). Further a strep-tag, fused to the C-terminal was used for purification (Hober et al, 2021). The beta version (B1.351) of the spike was produced with three mutations in the RBD part (K417N E484K N501Y).

## Peptides

A total of 42 20-mer peptides with 10 aa overlap, corresponding to the huCoV-19/WH01 RBD (25 peptides), M (22 peptides), and N (41 peptides) and Bat N (42 peptides) were purchased from Sigma-Aldrich (St. Louis, MO). The peptides were divided in pools of 4–5 or 8–10 peptides/pool depending on experimental setup.

## Immunization and infection schedules in mice and rabbits

BALB/c (H-2$^d$) or C57BL/6 (H-2$^b$) ($n = 5$) mice were immunized up to three times with 3-week intervals, and sacrificed 2 weeks after the last immunization for spleen and blood collection as described previously (Ahlén et al, 2020; Maravelia et al, 2021). Twenty K18-hACE2 mice were divided into four groups ($n = 5$) and immunized with indicated vaccines. Each K18-hACE mouse received three immunizations with 3 weeks between each injection. Two weeks after the last immunization, the K18-hACE2 mice were infected with SARS-CoV-2 Beta. Immunization method in brief, BALB/c or K18-hACE2 mice (five per group) were immunized intramuscularly in the *Tibialis cranialis anterior* muscle with 50 μg plasmid DNA in a volume of 50 μl sterile phosphate-buffered saline (PBS) by regular needle (27G) injection followed by *in vivo* electroporation using a Cliniporator2 device (IGEA) using two needle electrodes. Prior to vaccine injections, mice were given analgesic and kept under isoflurane anesthesia during the vaccinations. During *in vivo* electroporation (in both mice and rabbits), a 1-ms 600-V/cm pulse followed by a 400-ms 60-V/cm pulse pattern was used to facilitate better uptake of the DNA. In addition, groups of mice were injected subcutaneously at the base of the tail with recombinant SARS-CoV-2 spike (S) or nucleo (N) protein mixed (1:1) with QS21 adjuvant (GMP grade, Alpha diagnostics).

Each New Zealand white rabbit was immunized with 84 or 840 μg of OC-2.4 DNA vaccine or only formulation buffer (Tris-EDTA, pH 7.4) vehicle. Injection were administered in the right Quadriceps muscle in 500 μl followed by *in vivo* electroporation under anesthesia using the GeneDriVe (IGEA) device and a 4 needle electrode array at a depth of 21 mm.

## Mouse challenge model

Two weeks post the last immunization, the K18-hACE2 mice were challenged with $1 \times 10^5$ pfu SARS-CoV-2 Beta variant via intranasal administration in a volume of 40 μl in a BSL-3 facility. The health of the animals was assessed daily for up to 13 days and evaluated based on several parameters, including body weight, general condition, piloerection, as well as movement and posture. At the time of euthanization, blood, nasal lavage, lungs, and spleen were collected.

## Detection of IgG-specific antibodies

Serum from mice and rabbits were used for detection of immunoglobulins against S or N protein. In brief, plates were coated with 1 μg/ml of recombinant S or N protein (Genescript) in 50 mM of Sodium Carbonate buffer pH 9.6 overnight at 4°C. Plates were blocked by incubation with dilution buffer (phosphate-buffered saline, 2% goat serum, 1% BSA) for 1 h at 37°C. Serum was added in serial dilutions with a starting dilution of 1:60 and then in serial dilution of 1:6. Serum antibodies were detected by an alkaline phosphatase conjugated goat anti-mouse IgG (Sigma A1047) 1:1,000 or mouse anti-rabbit IgG (Sigma A2556) 1:1,000 and visualized using p-nitrophenyl phosphate substrate solution. Optical density (OD) was read at 405 nm with a 620 nm background. Antibody titers were determined as endpoint serum dilutions at which the OD value was at least three times the OD of the negative control (nonimmunized or control animal serum) at the same dilution.

## Detection of specific IFN-γ producing T cells and antibodies

Two weeks post last vaccination, splenocytes from each group of wild-type mice or peripherial blood mononuclear cells (PBMCs) from rabbits were harvested and tested for their ability to induce specific T cells based on IFN-γ secretion after peptide or protein stimulation for 48 h essentially as described (Hawman et al, 2021; Maravelia et al, 2021) using a commercially available enzyme-linked immunospot (ELISpot) assay (Mabtech).

## Virus propagation

The SARS-CoV-2 huCoV-19/WH01, Beta, and Omicron strains were isolated from patient samples at the Public Health Agency of Sweden and confirmed by sequencing. The SARS-CoV-2 Delta variant was provided by Dr. Charlotta Polacek Strandh, Statens Serum Institute, Copenhagen, Denmark. All variants were propagated on Vero-E6 cells and titered using a plaque assay as previously described (Varnaitė et al, 2020), with fixation after 72 h. The huCoV-19/WH01, Delta, and Omicron strains used in this study was passaged three times and the Beta strain two times. All infectious experiments were performed in a BSL-3 facility approved by the Swedish Board of Work Environment Safety.

## Neutralization of SARS-CoV-2 *in vitro*

Titer of neutralizing antibodies in serum from mice and rabbits were determined by CPE-based microneutralization assay. For mice, serum from each vaccination group was pooled, while for rabbits serum from each individual was tested. Briefly, serum was heat inactivated at 56°C for 30 min before serial diluted 2-fold. Each dilution was conducted in quadruplets and mixed with 500 pfu of SARS-CoV-2 huCoV-19/WH01, Beta, Delta, or Omicron in a 1:1 dilution. After 1 h of incubation at 37°C, 5% $CO_2$ 100 μl of serum–virus mix were added to Vero E6 cells on a 96-well plate ($20 \times 10^4$ cells/well) and incubated for 72 h at 37°C, 5% $CO_2$. CPE for each well was determined using a Nicon Eclipse TE300 microscope. As controls, wells with medium only, diluted serum only, virus only, and serum known to contain SARS-CoV-2 neutralizing antibodies mixed with virus were included in each experiment. All infectious experiments were performed in a BSL-3 facility approved by the Swedish Board of Work Environment Safety.

### PCR/viral RNA

Trizol (Sigma-Aldrich) in a ratio of 1:3 was used to inactivate potential virus in nasal lavage samples (50 μl) from SARS-CoV-2-infected K18-hACE2 mice. For lung and spleen, PBS was added to each sample (1 g/ml) and pestles were used to crush the organs. Thereafter, the samples were centrifuged (5 min at 7,000 rpm) and 50 μl of each lung or spleen sample was added to Trizol (1:3). Total RNA was extracted using the Direct-zol RNA Miniprep kit (Zymo Research) according to the manufacturer's instructions. Viral RNA were thereafter measured by quantitative real-time polymerase chain reaction (qRT-PCR) using TaqMan Fast Virus 1-Step master mix (Thermo Fisher Scientific) with primers and probe for the SARS-CoV-2 E gene.

> Forward: 5′- ACAGGTACGTTAATAGTTAATAGCGT -3′
> Reverse: 5′- ATATTGCAGCAGTACGCACACA -3′
> Probe: FAM- ACACTA GCC ATC CTT ACT GCG CTT CG MGB

For lung and spleen samples, mouse ACTB mix (Thermo Fisher Scientific) was used as endogenous control. The PCR reaction was performed using a capillary Roche LightCycler 2.0 system.

### Histological analysis

The lungs were formalin fixed, embedded in paraffin, and sectioned for H&E staining. The sections were analyzed by an independent veterinary pathologist blinded to the treatment groups. All sections were scored according to bronchial and alveolar signs of inflammation and disease.

### Statistical analysis

Data were analyzed with use of on GraphPad Prism V.5 software and Microsoft Excel V.16.13.1. The group size of animals were kept to a minimum (5 female mice or 3 + 3 male and female rabbits) to allow for a meaningful statistical analysis size. All experiments included replicate determinations of three or five replicates, which relates to both the number of animals and the number of replicates in the *in vitro* assay.

# Data availability

All data presented in the manuscript are publicly accessible at https://ki.se/en/research/opencorona.

**Expanded View** for this article is available online.

## Acknowledgements

Our work was funded and supported by the OPENCORONA consortium that has received funding from the European Union's Horizon 2020 research and innovation program under grant agreement no. 101003666. The study was also supported by The Swedish Research Council, The Swedish Cancer Society, Vinnova project CAMP (Contract no. 2017-02130), Horizon 2020 ERINHA network grant, Knut and Alice Wallenberg foundation, Science for Life Laboratory (SciLifeLab), Erling-Persson family foundation, and by private donations to MS and AM. Karolinska Institutet supported LF.

---

**The paper explained**

**Problem**
The SARS-CoV-2 pandemic has been effectively slowed down by the extensive use of various vaccines. However, the ability of the virus to change its genetic set-up poses a challenge. Thus, new types of vaccines targeting regions with lower genetic variability are needed.

**Results**
We generated a completely new DNA vaccine design that included genes encoding three unique loops of the receptor-binding domain of the spike protein, corresponding to three variants of SARS-CoV-2 (WH1, Alpha, and Beta). In addition, the genes for both the membrane and the nucleoproteins were included in the vaccine. We could show that the vaccine induce high level of antibodies that neutralized the SARS-CoV-2 WH1, Beta, Delta, and Omicron variants in cell culture. Importantly, the vaccine induced antibodies and T cells that protected mice expressing the human angiotensin-converting enzyme-2 receptor, used by SARS-CoV-2 for cellular entry, from lethal disease and that reduced viral replication. We could also show that the priming of nucleoprotein-specific T cells alone, had a partially protective effect against lethal disease.

**Impact**
We aim for taking the herein described vaccine to a first-in-man phase 1 clinical trial to evaluate the safety and immunogenicity of the vaccine as a booster vaccine to existing mRNA vaccines. If a booster dose with the current vaccine can broaden immune responses in humans vaccinated to existing mRNA vaccines, we hope that this response can better protect against new variants of SARS-CoV-2 appearing during the pandemic.

## Author contributions

**Matti Sällberg:** Conceptualization; resources; formal analysis; supervision; funding acquisition; investigation; methodology; writing – original draft; project administration. **Sofia Appelberg:** Data curation; formal analysis; methodology; writing – original draft; writing – review and editing. **Gustaf Ahlén:** Data curation; formal analysis; supervision; validation; methodology; writing – original draft; project administration; writing – review and editing. **Negin Nikouyan:** Data curation; methodology; writing – original draft; writing – review and editing. **Jingyi Yan:** Data curation; formal analysis; supervision; methodology; writing – original draft; writing – review and editing. **Sofie Weber:** Data curation; formal analysis; writing – original draft; writing – review and editing. **Olivia Larsson:** Formal analysis; supervision; writing – original draft; writing – review and editing. **Urban Höglund:** Formal analysis; supervision; writing – original draft; writing – review and editing. **Soo Aleman:** Formal analysis; writing – original draft; writing – review and editing. **Friedemann Weber:** Supervision; methodology; writing – original draft; writing – review and editing. **Emma Perlhamre:** Writing – original draft; project administration; writing – review and editing. **Johanna Apro:** Writing – original draft; project administration; writing – review and editing. **Eva-Karin Gidlund:** Validation; methodology; writing – original draft; writing – review and editing. **Ola Tuvesson:** Data curation; supervision; validation; methodology; writing – original draft; writing – review and editing. **Simona Salati:** Formal analysis; methodology; writing – original draft; writing – review and editing. **Matteo Cadossi:** Formal analysis; methodology; writing – original draft; writing – review and editing. **Hanna Tegel:** Formal analysis; methodology; writing – original draft; writing – review and editing. **Sophia Hober:** Formal analysis; methodology; writing – original draft; writing – review and editing. **Lars Frelin:** Conceptualization; formal analysis; supervision; methodology; writing –

original draft; project administration; writing – review and editing. **Ali Mirazimi:** Conceptualization; formal analysis; supervision; methodology; writing – original draft; project administration; writing – review and editing.

In addition to the CRediT author contributions listed above, the contributions in detail are:
The study was designed, analyzed, and written by MS, AM, FW, LF, and GA. SA, GA, LF, MS, JY, NN, and SW performed all experiments. The EPSGun was designed and built by MC and SS. Recombinant proteins (except N) were produced and purified by HT and SH. The toxicological study in rabbits was designed by MS, UH, OL, SooA, EP, JA, LF, SS, MC, and GA. The plasmid was produced and purified by E-KG and OT. All authors reviewed and edited the manuscript.

## Disclosure and competing interests statement

MS and LF are founders, owners, and paid consultants to Svenska Vaccinfabriken Produktion AB that holds patent applications related to the vaccine described herein. MC and SS are employees of IGEA Biomedical that owns the EPSgun and the GeneDrive. OT and E-KG are employees of Northx Biologics. UH, OL, and SW are employees of Adlego Biomedical AB.

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
