## [Review Process File · EMBO Molecular Medicine]

A universal SARS-CoV DNA vaccine inducing highly crossreactive neutralizing antibodies and T cells

Matti Sällberg, Sofia Appelberg, Gustaf Ahlen, Negin Nikouyan, Jingyi Yan, Sofie Weber, Olivia Larsson, Urban Höglund, Soo Aleman, Friedemann Weber, Emma Perlhamre, Johanna Apro, Eva-Karin Gidlund, Ola Tuveesson, Simona Salati, Matteo Cadossi, Hanna Tegel, Sophia Hober, Lars Frelin, and Ali Mirazimi

DOI: [10.15252/emmm.202215821](https://doi.org/10.15252/emmm.202215821)

Corresponding author: [Matti Sällberg \(matti.sallberg@ki.se\)](mailto:matti.sallberg@ki.se)

Review Timeline:

Submission Date:	4th Feb 22
Editorial Decision:	23rd Feb 22
Revision Received:	13th Jun 22
Editorial Decision:	27th Jun 22
Revision Received:	7th Jul 22
Editorial Decision:	22nd Jul 22
Revision Received:	16th Aug 22
Accepted:	17th Aug 22

Editor: *Zeljko Durdevic*

Transaction Report:

23rd Feb 2022

Dear Dr. Sällberg,

Thank you for the submission of your manuscript to EMBO Molecular Medicine. We have now received feedback from the three reviewers who agreed to evaluate your manuscript. As you will see from the reports below, while the referee #3 is supportive of the study, referees #1 and #2 recognize the interest of the study but also raise serious concerns that should be addressed in a major revision.

We would welcome the submission of a revised version within three months for further consideration. Please let us know if you require longer to complete the revision.

Please use this link to login to the manuscript system and submit your revision: Link Not Available

I look forward to receiving your revised manuscript.

Yours sincerely,

Zeljko Durdevic

***** Reviewer's comments *****

Referee #1 (Remarks for Author):

In their manuscript entitled „Universal SARS-CoV DNA vaccine inducing highly crossreactive neutralizing antibodies and T cells" Sofia Appelberg and coworker describe an interesting approach for the development of a more universal Sars-CoV vaccine. Their concept is based on a DNA-vector encoding a fusion protein encompassing the RBD domains of the wt, the beta and alpha variant and the M and N protein. After immunization of mice the authors observe the induction of neutralizing antibodies. In challenge experiments immunized mice were protected from lethal infection.

There are some open points to be addressed.

The authors should show the expression of the OC2.4 construct in transfected cells by immunofluorescence microscopy or Western blot analysis.

The analysis of the T cell response is central for the manuscript. The data should be transferred from the supplement into the manuscript

The structure of the fusion protein is not fully clear. Is the M protein separated by a P2A sequence from the N protein as described in the text?

The titer of S-binding antibodies should be analyzed after the first, second and third vaccination.

If the authors have data describing the Th1 /Th2 polarization these data should be included in the supplement

Referee #2 (Remarks for Author):

Appelberg and colleagues determined the immunogenicity and efficacy of a DNA construct containing receptor binding domain (RBD) loops of the S protein corresponding to the huCoV-19/WH01, Alpha and Beta variants, combined with the M and N proteins of the huCoV-19/WH01 variant to target broadest immune responses. Despite demonstrating some protection against beta virus challenge in mice, there is no evidence that this vaccine is equal or better than licensed vaccines. The main concerns are below:

1. The authors refer to SARS-CoV in the title and text and propose that this approach will give broad reactivity across beta coronaviruses. However, it seems that the vaccine construct contains only genetic elements of SARS-CoV-2, and studies to prove that cross-reactivity or functionality with SARS-CoV or other beta coronaviruses are actually improving the vaccine efficacy are missing. The "SARS-CoV OC-2.4" DNA construct is not clearly described and the author refers to Dai et al Cell 2020 which report a recombinant bivalent RBD approach that does not appear to be relevant for this study.
2. In terms of humoral immunity, the construct does not provide a significant improvement compared to recombinant S protein or spike-DNA.
3. Challenge model with wuhan strain to compare S protein vs DNA is missing, and the challenge model with beta strain does not demonstrate superiority of the DNA construct compared to recombinant S protein. The rationale of use of recombinant S protein + QS21 is not clearly described in the text.
4. T cell cross-reactivity with bat coronavirus is not relevant for the study. Moreover, cross-reactivity levels (figure S3) are below threshold of detection for 4 mice out of 6.
5. Evidence of 60% protection with N protein immunization is not supported histological and RNA evidence, with worsening of clinical score in animals post challenge compared to other treatments. Moreover, this is not relevant for the DNA vaccine.
6. Statistical values, experimental replicates and description of experimental conditions are often missing in the figures and text.
7. There are plenty of typos in the title, abstract, text and figures that makes it difficult to understand the manuscript.

Overall, the manuscript is difficult to read and the conclusions are not supported by relevant experimental evidence. Despite the

aim of expanding breadth of response, additional work is necessary to clarify rationale of the study, to improve the DNA construct and to generate the right controls and the missing data. Thus, I believe the manuscript is not suitable for publication.

Referee #3 (Remarks for Author):

In this elegant study, the group of Matti Sallberg from Sweden describes a universal SARS-CoV DNA vaccine that induces highly cross-reactive neutralizing antibodies and T cells. This group is leading in vaccine development and the data shown convincing. The transgenic moused model used supports the protective role of the induced immune response and also supports a critical role of T cells as has been suggested in the human system. Overall, in this reviewer's opinion, this is a well performed study that addresses an important and highly relevant topic.

I have only one comment:

The authors use different vaccine strategies, the rationale for which should be discussed or at least be explained.

Replies to comments made by the reviewers.

Referee #1 (Remarks for Author):

In their manuscript entitled „Universal SARS-CoV DNA vaccine inducing highly crossreactive neutralizing antibodies and T cells" Sofia Appelberg and coworker describe an interesting approach for the development of a more universal Sars-CoV vaccine. Their concept is based on a DNA-vector encoding a fusion protein encompassing the RBD domains of the wt, the beta and alpha variant and the M and N protein. After immunization of mice the authors observe the induction of neutralizing antibodies. In challenge experiments immunized mice were protected from lethal infection.

There are some open points to be addressed.

*The authors should show the expression of the OC2.4 construct in transfected cells by immunofluorescence microscopy or Western blot analysis.

REPLY: A western blot showing detection of the M-N fusion protein has been enclosed as supplementary figure 1.

*The analysis of the T cell response is central for the manuscript. The data should be transferred from the supplement into the manuscript

REPLY: The T cell data has been included in Figure 2. Also, more data has been included.

*The structure of the fusion protein is not fully clear. Is the M protein separated by a P2A sequence from the N protein as described in the text?

REPLY: The P2A sequence is only present between the RBD and M sequence. The M and N proteins are present as a fusion protein. This has now been clarified in the text, and has been shown in western blot in Supplemental Figure 1.

*The titer of S-binding antibodies should be analyzed after the first, second and third vaccination.

REPLY: This has now been included in Supplemental Figure 2.

*If the authors have data describing the Th1 /Th2 polarization these data should be included in the supplement

REPLY: We have focused on the cross reactivity of the T cells, also the strong IFN γ responses certainly suggest a Th1-like response. This will however be included in the next publication.

Referee #2 (Remarks for Author):

Appelberg and colleagues determined the immunogenicity and efficacy of a DNA construct containing receptor binding domain (RBD) loops of the S protein corresponding to the huCoV-19/WH01, Alpha and Beta variants, combined with the M and N proteins of the huCoV-19/WH01 variant to target broadest immune responses. Despite demonstrating some protection against beta virus challenge in mice, there is no evidence that this vaccine is equal or better than licensed vaccines. The main concerns are below:

1. The authors refer to SARS-CoV in the title and text and propose that this approach will give broad reactivity across beta coronaviruses. However, it seems that the vaccine construct contains only genetic elements of SARS-CoV-2, and studies to prove that cross-reactivity or functionality with SARS-CoV or other beta coronaviruses are actually improving the vaccine efficacy are missing. The "SARS-CoV OC-2.4" DNA construct is not clearly described and the author refers to Dai et al Cell 2020 which report a recombinant bivalent RBD approach that does not appear to be relevant for this study.

REPLY: Thank you for the comment. The description of the SARS-CoV OC-2.4 vaccine has now been better described. We agree that much of the data provided show cross reactivity between SARS-CoV-2 variants. However, we clearly show that vaccine primed T cells are cross reactive with Bat-SARS-CoV. Hence, we feel that it is correct to state that the OC-2.4 vaccine induces immune responses cross reactive with SARS-CoVs.

*2. In terms of humoral immunity, the construct does not provide a significant improvement compared to recombinant S protein or spike-DNA.

REPLY: This is correct. However, the major point of the current vaccine design is to activate both humoral and broadly reactive T cells. Thus, we feel that it is quite impressive that the anti-S responses are comparable to S-based vaccines, but with the addition that new responses to M and N are induced.

*3. Challenge model with wuhan strain to compare S protein vs DNA is missing, and the challenge model with beta strain does not demonstrate superiority of the DNA construct compared to recombinant S protein. The rationale of use of recombinant S protein + QS21 is not clearly described in the text.

REPLY: We feel that the most relevant control is the Beta variant. Also, we agree that the vaccine does not show superiority in the *in vivo* model. However, it is important to stress that we do induce new T cell specificities to highly conserved parts of the virus, such as M and N. This suggests, together with the T cell data, that we have a vaccine that complements the existing vaccines by adding new T cell

repertoires. Importantly, as we show, the activation of N-specific T cells alone has protective properties, supporting the concept.

*4. T cell cross-reactivity with bat coronavirus is not relevant for the study. Moreover, cross-reactivity levels (figure S3) are below threshold of detection for 4 mice out of 6.

REPLY: We disagree with the reviewer's comment. The cross reactivity with Bat N sequences is central to a universal SARS-CoV vaccine, since new viruses are likely to appear from animals such as bats. Thus, even if the S gene is replaced, a prior SARS-CoV infection, or vaccination using our type of vaccine, will still have the chance of offering some degree of protection. Indeed, that is what the experiment with vaccination with N protein alone shows. Thus, the cross reactivity of the vaccine-primed T cells to Bat N sequences is highly relevant.

*5. Evidence of 60% protection with N protein immunization is not supported histological and RNA evidence, with worsening of clinical score in animals post challenge compared to other treatments. Moreover, this is not relevant for the DNA vaccine.

REPLY: We disagree with the reviewer as we feel that the data showing that priming T cells only has a partially protective effect is highly relevant. This is a novel observation that extends data from humans regarding the role of T cells. Although there is histological signs of disease the increased survival clearly shows benefit of priming N specific T cells. Thus, although this was a protein-based vaccination it supports the concept of a protective role of cross-reactive T cells. Importantly, the N protein was from the WH1 strain and protected against a challenge with a Beta strain.

*6. Statistical values, experimental replicates and description of experimental conditions are often missing in the figures and text.

REPLY: We have included statistical analysis when it has been possible to do so. The experimental design has been clarified.

*7. There are plenty of typos in the title, abstract, text and figures that makes it difficult to understand the manuscript.

REPLY: We have corrected the text throughout the manuscript.

Overall, the manuscript is difficult to read and the conclusions are not supported by relevant experimental evidence. Despite the aim of expanding breadth of response, additional work is necessary to clarify rationale of the study, to improve the DNA

construct and to generate the right controls and the missing data. Thus, I believe the manuscript is not suitable for publication.

Referee #3 (Remarks for Author):

In this elegant study, the group of Matti Sallberg from Sweden describes a universal SARS-CoV DNA vaccine that induces highly cross-reactive neutralizing antibodies and T cells. This group is leading in vaccine development and the data shown convincing. The transgenic moused model used supports the protective role of the induced immune response and also supports a critical role of T cells as has been suggested in the human system. Overall, in this reviewers opinion, this is a well performed study that addresses an important and highly relevant topic.

I have only one comment:

The authors use different vaccine strategies, the rationale for which should be discussed or at least be explained.

REPLY: This has now been explained in the text.

27th Jun 2022

Dear Dr. Mirazimi,

Thank you for the submission of your revised manuscript to EMBO Molecular Medicine. We have now heard back from the two referees who we asked to re-evaluate your manuscript. As you will see from the reports below, while the referee #1 supports publication of your manuscript, referee #2 acknowledges the improvements of the revised manuscript but also raises a number of concerns that should be addressed in an additional and final round of major revision. Only one point considering a control group immunized with N protein + QS21 requires additional experimentation, while all the other points should be addressed by additional clarifications and discussion.

Further consideration of a revision that addresses reviewer's concerns in full will entail an additional round of review. Acceptance or rejection of the manuscript will depend on the completeness of your responses included in the next, final version of the manuscript. For this reason, and to save you from any frustrations in the end, I would strongly advise against returning an incomplete revision.

We would welcome the submission of a revised version within three months for further consideration. Please let us know if you require longer to complete the revision.

Please use this link to login to the manuscript system and submit your revision: Link Not Available

I look forward to receiving your revised manuscript.

Yours sincerely,

Zeljko Durdevic

**** Reviewer's comments ****

Referee #1 (Comments on Novelty/Model System for Author):

The authors adequately addressed all points raised in the previous review.

Referee #1 (Remarks for Author):

The authors adequately addressed all points raised in the previous review. This is a very relevant, well designed and well performed study.

Referee #2 (Comments on Novelty/Model System for Author):

Please see suggested experiments and manuscript review in the comments to the authors.

Referee #2 (Remarks for Author):

The authors focus their discussion on the benefit of inducing N specific T cell response which are cross-reactive with other coronaviruses. When recombinant N protein + adjuvant is used, it results in 60% protection against challenge (Figure 3C). The DNA vaccine is reported to induce both N and spike immunity (Figure 2B,F,G), broadening the immune response, however no added protection is observed in the challenge model (recombinant Wh01 spike protein and DNA vaccine confer full protection against beta challenge, Figure 3C). The study has several limitations which the authors have highlighted in the new version and the flow is clearer. However, I believe the significance of this is limited, and it could be considered for publication provided additional comments are addressed.

Additional comments:

- Clarify why QS21 adjuvant was used
- Remove statements related N specific responses whose data is not shown
- Regarding Figure 2C, additional M and N specific T cell are induced after booster however no significance analysis is given, this should be stated in the text.
- Figure 2E-H should be clearly described in the results section, clarifying that specific IFN γ responses are below limit of detection for several mice/group.
- A control group in Figure 2 immunized with N protein + QS21 is missing, this is necessary to define the level of N specific immunity that correlates with in vivo protection, and to compare with DNA-induced immunity
- Clarify in figure legend and materials&methods what is 1-10 in x axis of Figure 2A-D
- Add p values asterisks in figure 3D

Replies to comments made by the reviewers.

Referee #1 (Comments on Novelty/Model System for Author):

The authors adequately addressed all points raised in the previous review.

Referee #1 (Remarks for Author):

The authors adequately addressed all points raised in the previous review. This is a very relevant, well designed and well performed study.

We thank the reviewer for the comment.

Referee #2 (Comments on Novelty/Model System for Author):

Please see suggested experiments and manuscript review in the comments to the authors.

Referee #2 (Remarks for Author):

The authors focus their discussion on the benefit of inducing N specific T cell response which are cross-reactive with other coronaviruses. When recombinant N protein + adjuvant is used, it results in 60% protection against challenge (Figure 3C). The DNA vaccine is reported to induce both N and spike immunity (Figure 2B,F,G), broadening the immune response, however no added protection is observed in the challenge model (recombinant Wh01 spike protein and DNA vaccine confer full protection against beta challenge, Figure 3C). The study has several limitations which the authors have highlighted in the new version and the flow is clearer. However, I believe the significance of this is limited, and it could be considered for publication provided additional comments are addressed.

Additional comments:

- Clarify why QS21 adjuvant was used

REPLY: The use of the QS21 adjuvant was based on the fact that it is commercially available for research and clinical use. Also, new Expanded View Figure 3 shows that QS21 is superior to alum, the today most widely used vaccine adjuvant in humans.

- Remove statements related N specific responses whose data is not shown

REPLY: We have now included data that support the statement regarding N antibodies on line 10, page 5.

- Regarding Figure 2C, additional M and N specific T cell are induced after booster however no significance analysis is given, this should be stated in the text.

REPLY: This has now been included in the text. Also, statistical comparisons have been included for both mice and rabbit T cell responses.

- Figure 2E-H should be clearly described in the results section, clarifying that specific IFN γ responses are below limit of detection for several mice/group.

REPLY: This has now been clarified in the text that some rabbits failed to develop any T cell responses at any time point.

- A control group in Figure 2 immunized with N protein + QS21 is missing, this is necessary to define the level of N specific immunity that correlates with in vivo protection, and to compare with DNA-induced immunity

REPLY: An additional experiment has been added to clarify this issue, Extended View 3. Also described in Results on page 10.

- Clarify in figure 1e legend and materials&methods what is 1-10 in x axis of Figure 2A-D

REPLY: Thank you for noticing this. This has now been added in the legend, that this corresponds to peptide pools.

- Add p values asterisks in figure 3D

REPLY: This has now been done.

22nd Jul 2022

Dear Prof. Mirazimi,

Thank you for the submission of your revised manuscript to EMBO Molecular Medicine. I am pleased to inform you that we will be able to accept your manuscript pending the following final amendments:

1) Authorship: In our submission system Ali Mirazimi and Matti Sällberg are indicated as corresponding authors, while in the manuscript only Matti Sällberg is the corresponding author. Please clarify and correct either in the submission system or in the manuscript.

2) E-mail addresses: Please update e-mail addresses for Negin Nikoyan, Johanna Apro, Emma Perlhamre, Ola Tuveson, Olivia Larsson.

3) In the main manuscript file, please do the following:

- Correct/answer the track changes suggested by our data editors by working from the attached/uploaded document.
- Add up to 5 keywords.
- Remove font colour.
- Remove data not shown (p.6).
- Add callouts for the figures 1H and 2G-H.
- Move M&M section after Discussion.
- In M&M, please specify the biosafety level for the experiments with SARS-CoV-2 by adding and amending the following sentence: All experiments with SARS-CoV-2 were performed in a ... level laboratory and with approval from...
- In M&M, statistical paragraph should reflect all information that you have filled in the Authors Checklist, especially regarding randomization, blinding, replication.
- Data availability statement should contain information about data that cannot be published in the manuscript itself (e.g. structural data, high-throughput sequencing or data from large-scale gene expression experiments). If no data are deposited in public repositories, please add the sentence: "This study includes no data deposited in external repositories".

Please check "Author Guidelines" for more information.

<https://www.embopress.org/page/journal/17574684/authorguide#availabilityofpublishedmaterial>

- Add legend for the figure 2H.

- Please rename "Conflict of Interest" to "Disclosure Statement & Competing Interests". We updated our journal's competing interests policy in January 2022 and request authors to consider both actual and perceived competing interests. Please review the policy <https://www.embopress.org/competing-interests> and update your competing interests if necessary.

- Correct the reference citation in the text and reference list. In the text, a reference should be cited by author and year of publication. Include a space between a word and the opening parenthesis of the reference that follows. In the reference list, citations should be listed in alphabetical order. Where there are more than 10 authors on a paper, 10 will be listed, followed by "et al.". Please check "Author Guidelines" for more information.

<https://www.embopress.org/page/journal/17574684/authorguide#referencesformat>

4) Supplementary table: Please upload the table as .xlsx or .doc file and rename it EV Table 1. Also, update the callouts in the main text.

5) Author Contributions: CRediT has replaced the traditional author contributions section because it offers a systematic machine readable author contributions format that allows for more effective research assessment. You are encouraged to use the free text boxes beneath each contributing author's name to add specific details on the author's contribution. More information is available in our guide to authors.

6) Authors Checklist: Please upload the .xlsx file.

7) Synopsis:

- Synopsis image: Please provide a visual abstract as a high-resolution .jpeg file 550 px-wide x (250-400)-px high.

- Synopsis text: Besides the short standfirst, please include 2-5 one sentence bullet points that summarise the paper as a .doc file. Please write the bullet points to summarise the key NEW findings. They should be designed to be complementary to the abstract - i.e. not repeat the same text. We encourage inclusion of key acronyms and quantitative information (maximum of 30 words / bullet point). Please use the passive voice.

8) For more information: There is space at the end of each article to list relevant web links for further consultation by our readers. Could you identify some relevant ones and provide such information as well? Some examples are patient associations, relevant databases, OMIM/proteins/genes links, author's websites, etc...

9) Press release: Please inform us as soon as possible and latest at the time of submission of the revised manuscript if you plan a press release for your article so that our publisher could coordinate publication accordingly.

10) Please be aware that we use a unique publishing workflow for COVID-19 papers: a non-typeset PDF of the accepted manuscript is published as "Just Accepted" on our website. With respect to a possible press release, we have the option to not post the "Just Accepted" version if you prefer to wait with the press release for the typeset version. Please let us know whether you agree to publication of a "Just accepted" version or you prefer to wait for the typeset version.

11) As part of the EMBO Publications transparent editorial process initiative (see our Editorial at <http://embomolmed.embopress.org/content/2/9/329>), EMBO Molecular Medicine will publish online a Review Process File (RPF) to accompany accepted manuscripts. This file will be published in conjunction with your paper and will include the anonymous referee reports, your point-by-point response and all pertinent correspondence relating to the manuscript. Let us know whether you agree with the publication of the RPF and as here, if you want to remove or not any figures from it prior to publication. Please note that the Authors checklist will be published at the end of the RPF.

12) Please provide a point-by-point letter INCLUDING my comments as well as the reviewer's reports and your detailed responses (as Word file).

I look forward to reading a new revised version of your manuscript as soon as possible.

Yours sincerely,

Zeljko Durdevic

***** Reviewer's comments *****

Referee #2 (Remarks for Author):

The authors addressed most of the concerns on the manuscript adding supplementary experimental data and adding more details in the main text. I am ok for publication.

The authors performed the requested editorial changes.

We are pleased to inform you that your manuscript is accepted for publication and is now being sent to our publisher to be included in the next available issue of EMBO Molecular Medicine.